# A Robust Parallel Initialization Method for Monocular Visual-Inertial SLAM

**DOI:** 10.3390/s22218307

**Published:** 2022-10-29

**Authors:** Min Zhong, Yiqing Yao, Xiaosu Xu, Hongyu Wei

**Affiliations:** Key Laboratory of Micro-Inertial Instruments and Advanced Navigation Technology, Ministry of Education, School of Instrument Science and Engineering, Southeast University, Nanjing 210096, China

**Keywords:** SLAM, initialization, visual-inertial, monocular

## Abstract

In order to improve the initialization robustness of visual inertial SLAM, the complementarity of the optical flow method and the feature-based method can be used in vision data processing. The parallel initialization method is proposed, where the optical flow inertial initialization and the monocular feature-based initialization are carried out at the same time. After the initializations, the state estimation results are jointly optimized by bundle adjustment. The proposed method retains more mapping information, and correspondingly is more adaptable to the initialization scene. It is found that the initialization map constructed by the proposed method features a comparable accuracy to the one constructed by ORB-SLAM3 in monocular inertial mode. Since the online extrinsic parameter estimation can be realized by the proposed method, it is considered better than ORB-SLAM3 in the aspect of portability. By the experiments performed on the benchmark dataset EuRoC, the effectiveness and robustness of the proposed method are validated.

## 1. Introduction

In the research field of multi-sensor fusion simultaneous localization and mapping (SLAM), the fusion of vision and inertia has always been a hot research topic in recent years [1,2,3,4,5,6,7,8]. The rich environmental information captured by image can be used for building maps and estimating the up-to-scale trajectory of the camera, which is attached to the mobile platform. The measurements of gyroscope and accelerometer contained in the inertial measurement unit (IMU) sensor can be integrated to estimate accurate short-term rigid body motion. In addition, both the camera and the IMU have matured miniaturized products. These properties make the combination of the monocular vision camera and the IMU economical, effective, and low energy for localization and mapping in small mobile robot or other platforms with limited load. 

The state-of-the-art visual-inertial SLAM methods [1,2,3,4,5,6,7,8] presented in the literature widely use the IMU preintegration [9] and the keyframe-based nonlinear optimization methods [6]. However, the performance of these techniques heavily depends on prior precise extrinsic calibration of the six-degree-of-freedom (6-DOF) transformation between the camera and the IMU. As the coordinate conversion parameters between the camera and the IMU, the extrinsic parameters play important roles in accurate data fusion between the camera and the IMU. Incorrect extrinsic calibration leads to system deviation and degrades the overall navigation performance. For the acquisition of extrinsic parameters, most of the visual-inertial SLAM methods [1,2,3,4,5,6,7] use Kalibr to calibrate the extrinsic parameters, which a state-of-the-art marker-based off-line calibration method [10]. Thereafter, the calibrated extrinsic parameters by Kalibr are considered accurate and constant during the running of the whole program. However, this process is complex and time consuming. Additionally, it is usually required to repeat this process when the sensors are repositioned. Therefore, in order to improve the portability and operability of the program, an automatic extrinsic parameter estimation is necessary. At present, the automatic extrinsic parameter calibration [11] has been implemented in a monocular visual-inertial system constructed by one camera and a low-cost IMU (VINS-Mono) [8]. Weibo Huang et al. [12] implemented self-calibration of the camera and the IMU extrinsic parameters in the monocular ORB feature-based SLAM system (ORB-SLAM) framework [1,2]; ORB are binary features invariant to rotation and scale (in a certain range), resulting in a very fast recognizer with good invariance to viewpoint, although this code is not open source.

For visual measurement processing, it can be divided into the optical flow methods [13] and the feature-based methods [14,15,16] according to the matching strategy.

The feature-based methods represented by ORB-SLAM series [1,2,3,4] extract the ORB feature points [14] from the image, then calculate the descriptor of the corresponding feature points, and finally obtain the feature point pairs in two frames through the descriptor comparison. This kind of method features high positioning accuracy, but it is time-consuming and easy to lose tracking in a weak texture environment or due to blurred image in fast motion [17].

The same as feature-based methods, the optical flow methods represented by VINS-Mono also extract feature points from the image. However, the latter ones use image pixel values to match feature points, and eliminate the descriptor calculation of feature points, which is the most time-consuming process in feature-based methods. This characteristic makes the optical flow methods suitable for real-time and resource-constrained occasions, while the accuracy is also guaranteed to a certain extent. Moreover, the robustness to pixel noise of the optical flow methods is better than feature-based methods. 

It must be noted that, due to the assumption of photometric invariance [14], the optical flow methods are easily affected by external illumination, which means the feature-based methods feature a better robustness to external illumination than optical flow methods. Hence, a mixed method can be built based on the optical flow and feature-based methods, in order to achieve their robustness complementarity. The optical flow methods complement the feature point method when the image descriptor cannot be accurately calculated due to motion blur. The feature point methods provide better inter-frame pose results to suppress the motion drift caused by the wrong feature matching of the optical flow method. 

Although the robustness complementarity has been achieved in the mixed method, the risk of tracking loss in weak illumination and low texture environment cannot be eliminated. Thus, the participation of IMU is very necessary. 

The involvement of IMU data needs to go through an initialization process [18,19], which mainly completes the work of finding the gravity direction and correcting the bias of gyroscopes and accelerometers [20]. The initialization of IMU data has been implemented by many algorithms [1,2,3,4,5,6,7,8,9]. Nowadays, most of the state-of-the-art methods use the visual initialization information as a reference to correct the gyroscope and accelerometer bias of IMU [4,5,6,7,8] at the start. Taking VINS-Mono as an example [19], the initialization of the IMU is realized by aligning the IMU preintegration results with the vision-only structure from motion (SfM) results, which can obtain a rough estimation of IMU state vector. Then, the related parameters of the IMU are continuously optimized in the subsequent tracking process. This leads to the fact that the accuracy of VINS-Mono may not be ideal when the map is just initialized. For the monocular mode of ORB-SLAM2 [21], only two keyframes with sufficient matching feature points are needed for initialization, and the combination with IMU is not considered, so the initialization process is faster than that of VINS-Mono, but the map is constructed up-to-scale. Although the open-source code of ORB-SLAM3 [22] adds the mode of monocular camera with IMU, it also includes multi-map management and feature-based relocation and loop closing. The implementation of these functions relies on the continuous increase of feature saving and management with the expansion of the map range, which consumes a large amount of computer memory, making ORB-SLAM3 unable to run in real time on resource-constrained platforms. In addition, the self-calibration of extrinsic parameters is not considered in ORB-SLAM3.

Based on the above analysis, this paper proposes a robust parallel initialization method for monocular visual inertial SLAM, which integrates the characteristics of VINS-Mono and monocular mode of ORB-SLAM2 in the initialization phase. Based on the ORB-SLAM2 initialization algorithm framework and the VINS-Mono initialization algorithm framework, the research work of this paper first executes the monocular initialization thread in ORB-SLAM2 and the monocular inertial initialization process in VINS-Mono in parallel, and then uses the Umeyama algorithm [23] and bundle adjustment (BA) [24] optimization to integrate their initialization data. The result of this method is more robust, and with higher accuracy than the original VINS-Mono initialization. Compared with the original monocular initialization trajectory of ORB-SLAM2, the trajectory obtained by the initialization in this paper is in metric unit instead of up-to-scale, which better describes the real external environment. It also retains the online extrinsic parameter calibration function in VINS-Mono initialization, which improves the portability of the initialization algorithm. In the subsequent experimental comparison with the state-of-the-art ORB-SLAM3 initialization results, it can be found that the accuracy of the map obtained by the two initializations is comparable. The contributions can be summarized as follows:The design of two-thread parallel initialization makes SLAM system have better portability and initialization robustness.The designed initialization mode improves the initialization accuracy of the original VINS-Mono, which can provide a more accurate initial value of state estimation to the visual inertial SLAM system.The initialization state estimated by the proposed method makes the trajectory accurate like the one estimated by ORB-SLAM3 in monocular-inertial mode.

The remainder of this article is arranged as follows: In Section 2, the measurement preprocessing of the IMU and the camera information are described. In Section 3, the detail of the initialization process in the proposed method is introduced. Section 4 presents the experimental results of the improved initialization of monocular visual-inertial SLAM. Finally, the conclusions of this study are presented in Section 5.

## 2. Measurement Preprocessing

To implement the initialization method proposed in this paper, monocular camera and IMU measurements need to be preprocessed. For monocular visual measurements, we use the combination of the optical flow method and the feature-based method to track features between consecutive frames and detect new features in the latest frame. For IMU measurements, the outputs of IMU are pre-integrated between consecutive frames. 

### 2.1. Vision Preprocessing

The optical flow tracking algorithm and ORB feature-based method are carried out in parallel for each new image. In both of them, keyframes are selected and saved according to the tracking effect. The optical flow method judges keyframes according to the average disparity between the current frame and the previous keyframe. The feature-based method is judged by the number of tracked feature points. 

#### 2.1.1. Optical Flow Tracking Algorithm

The main function of the optical flow tracking algorithm in this paper is to match the feature points between successive image frames based on the optical flow method for the feature points extracted from image data, and to get the feature matching relationship between image frames.

Specifically, for each new frame, new corner feature points were detected after existing feature points were tracked by Kanade–Lucas–Tomasi (KLT) sparse optical flow algorithm [25], and this operation is to ensure the number of feature points (100–300) detected in each frame are sufficient. Then, outlier rejection is performed using RANSAC [26] with a fundamental matrix model. 

KLT corner tracking algorithm is also known as LK tracking algorithm. It is a classic corner tracking algorithm. The algorithm assumes that the target is in the video stream, only a small consistent displacement is generated, and the gray level of the target does not change much. The detailed description of the LK tracking algorithm is attached in Appendix A.

#### 2.1.2. ORB Feature-Based Method

The feature-based method includes detecting and matching steps of feature points. The form of feature points adopts ORB feature points [14], which include FAST corner extraction and BRIEF descriptor calculation. 

The main role of the method based on ORB feature in this paper is to obtain the motion relationship between two frames as well. The method is to calculate the matching relationship of ORB feature points extracted between two frames, and then to establish the initialized map according to the motion relationship and the position information of feature points obtained by the principle of epipolar geometry. The matching relationship is achieved by calculating the Hamming distance of the ORB descriptor of the relevant feature points in the image.

For instance, when the similarity between the descriptor of feature point A in image frame 1 and that of feature point B in image frame 2 is greater than 90%, A and B are considered to be the same feature point, that is, the two points are successfully matched. Since the motion relationship between image frame 1 and image frame 2 is unknown during the initialization process, we usually match all feature points within a certain range of pixel coordinates of feature point A of image 2 to obtain feature point B that matches feature point A in image frame 1.

### 2.2. IMU Preprocessing

IMU senses the information of acceleration and angular velocity of motion and provides relative positioning information. The combination of IMU and monocular camera can provide an absolute scale reference of the real environment for the SLAM system. The derivation process of preintegration is consistent with subsection B of section IV in VINS-Mono [27].

#### 2.2.1. Model of Output

An IMU measures the rotation rate and the acceleration of the sensors with respect to an inertial frame. The measurements, namely ${\hat{a}}_{t}$ and ${\hat{\omega }}_{t}$, are affected by additive white noise n and a slowly varying sensor bias b. Considering accelerometer bias $b_{a}$, gyroscope bias $b_{\omega },$ and additive noise, the raw measurements of accelerometer and gyroscope, $\hat{a}$ and $\hat{\omega }$, are separately modeled as follows:(1)$$\begin{array}{l} {\hat{a}}_{t}{=a}_{t}{+b}_{a_{t}}{+R}_{w}^{t}g^{w}{+n}_{a},\\ {\hat{\omega }}_{t}{=\omega }_{t}{+b}_{\omega_{t}}{+n}_{\omega } \end{array}$$

The first items on the right side of (1) are the true value of accelerometer and gyroscope, which are all three-axis measurement information, as shown in (3), the second term is the bias, and the last term is the measurement noise. When the accelerometer is stationary, it is affected by the gravity acceleration value in the world coordinate system $g^{w}$.

The additive noise in acceleration and gyroscope measurements are assumed as Gaussian white noise, $n_{a}\sim N\left({0,\;\sigma }_{a}^{2}\right)$, $n_{\omega }\sim N\left({0,\;\sigma }_{\omega }^{2}\right)$. Acceleration bias and gyroscope bias are modeled as random walk, ${\dot{b}}_{a_{t}}{=n}_{b_{a}}$, ${\dot{b}}_{\omega_{t}}{=n}_{b_{\omega }}$, whose derivatives are Gaussian white noise, $n_{b_{a}}\sim N\left({0,\;\sigma }_{b_{a}}^{2}\right)$, $n_{b_{\omega }}\sim N\left({0,\;\sigma }_{b_{\omega }}^{2}\right)$. The *t* subscript in the corresponding parameter denotes the value of the parameter at time *t*. $R_{w}^{t}$ represents the rotation matrix from the world coordinate system to the current inertial coordinate system.

#### 2.2.2. IMU Preintegration

For two consecutive frames $b_{k}$ and $b_{k+1}$, there exists several inertial measurements in time interval ${[t}_{k}{,\;t}_{k+1}]$. Given the bias estimation, we integrate them in local frame $b_{k}$ as:(2)$$\begin{array}{l} \alpha_{b_{k+1}}^{b_{k}}=\iint_{t_{k}}^{t_{k+1}}R_{t}^{b_{k}}\left({\hat{a}}_{t}{-b}_{a_{t}}\right){\mathrm{dt}}^{2}\\ \beta_{b_{k+1}}^{b_{k}}=\int_{t_{k}}^{t_{k+1}}R_{t}^{b_{k}}\left({\hat{a}}_{t}{-b}_{a_{t}}\right)\mathrm{dt}\\ \gamma_{b_{k+1}}^{b_{k}}=\int_{t_{k}}^{t_{k+1}}\frac{1}{2}\Omega \left({\hat{\omega }}_{t}{-b}_{\omega_{t}}\right)\gamma_{t}^{b_{k}}\mathrm{dt} \end{array},$$
where $\gamma_{b_{k+1}}^{b_{k}}$ is the quaternion representation of angular velocity increment, among
(3)$$\Omega \left(\omega \right)=\left[\begin{array}{cc} {-\lfloor \omega \rfloor }_{\times } & \omega \\ -\omega^{T} & 0 \end{array}\right],\;\;{\left\lfloor \omega \right\rfloor }_{\times }=\left[\begin{array}{ccc} 0 & -\omega_{z} & \omega_{y}\\ \omega_{z} & 0 & -\omega_{x}\\ -\omega_{y} & \omega_{x} & 0 \end{array}\right],$$

According to the output of accelerometer and gyro in Equation (1), $\alpha_{b_{k+1}}^{b_{k}}$, $\beta_{b_{k+1}}^{b_{k}}$, and $\gamma_{b_{k+1}}^{b_{k}}$ are the velocity increment pre-integrated measurements, the position increment pre-integrated measurements, and the rotation amount pre-integrated measurements of the motion between consecutive frames $b_{k}$ and $b_{k+1}$, respectively. It can be seen that the preintegration term (2) can be solely with IMU measurements by taking $b_{k}$ as the reference frame given bias. $R_{t}^{b_{k}}$ is the rotation matrix from the inertial frame at time t to the inertial frame of $b_{k}$. 

#### 2.2.3. Bias Correction

Equation (2) is derived basing on the assumption that the bias of gyro and additive table is constant in the integration interval, while in fact the bias changes slowly in the output process of IMU. When the bias changes, if the measurement is still calculated according to Equation (2), the pre-integral measurements need to be recomputed, which is computationally expensive. To solve this problem, if the estimation of bias changes minorly, we adjust $\alpha_{b_{k+1}}^{b_{k}}$, $\beta_{b_{k+1}}^{b_{k}}$, and $\gamma_{b_{k+1}}^{b_{k}}$ by their first-order approximations with respect to the bias as: (4)$$\begin{array}{l} \alpha_{b_{k+1}}^{b_{k}}\approx {\hat{\alpha }}_{b_{k+1}}^{b_{k}}{+J}_{b_{a}}^{\alpha }{\delta b}_{a_{k}}{+J}_{b_{\omega }}^{\alpha }{\delta b}_{\omega_{k}}\\ \beta_{b_{k+1}}^{b_{k}}\approx {\hat{\beta }}_{b_{k+1}}^{b_{k}}+J_{b_{a}}^{\beta }{\delta b}_{a_{k}}+J_{b_{a}}^{\beta }{\delta b}_{\omega_{k}}\\ \gamma_{b_{k+1}}^{b_{k}}\approx {\hat{\gamma }}_{b_{k+1}}^{b_{k}}⊗\left[\begin{array}{c} 1\\ \frac{1}{2}J_{b_{\omega }}^{\gamma }{\delta b}_{\omega_{k}} \end{array}\right] \end{array}$$
where $J_{y}^{x}$ denotes the Jacobian matrix of y versus x; refer to VINS-Mono for specific definitions. At the beginning, $\alpha_{b_{k}}^{b_{k}}$ and $\beta_{b_{k+1}}^{b_{k}},$ are **0**, and $\gamma_{b_{k}}^{b_{k}}$ is identity quaternion. Moreover, ⊗ is a quaternion multiplication operator.

## 3. The Initialization Method Integrates Feature-Based Pose Information, Optical Flow Pose Information and Inertial Information

If we process the results of the optical flow method, the feature-based method, and IMU preintegration at the same time, the amount of data to be optimized is considerable, which requires high computer performance; therefore, we use parallel threads to initialize and a two-step optimization. In order to make full use of the existing open-source programs and reduce the amount of calculation of the programs at the same time, we first initialize the optical flow data and feature-based data separately, and then fuse the two initialization results to complete the combination of visual information as well as the inertial information. 

The specific implementation process is shown in Figure 1. Firstly, the initialization threads based on feature points and optical flow method are carried out in parallel. After the completion of the initialization of the two threads, the coarse alignment of the initial map is realized by using the Umeyama algorithm. Then, the local BA fusion of visual items is performed. Finally, the visual and inertial items in the two threads are fused to obtain the optimized initialization results. 

### 3.1. Initialization Framework for Monocular VIO in the Optical Flow Method

The optical flow monocular inertial initialization method adopted by VINS-Mono is advanced and widely used. It also includes the extrinsic parameters estimation [11] between the camera and the IMU, and the alignment of the world coordinate with the gravity direction. The procedure of initialization framework for VINS is shown in Figure 2.

Visual tracking based on optical flow method requires continuous frames and small movement distance. However, in order to obtain the initial map points, the matching feature points need to be triangulated, and the premise of triangulation is that there is a large enough displacement between the two image frames where the feature points are located. Therefore, the initialization based on the optical flow method tends to establish a sliding window of size 10 as a data buffer to save the image frame, and the saving unit is one image frame. The initialized operation will find two frames with appropriate disparity in the sliding window for triangulation and propagate the triangulation results to other frames in the sliding window based on the results of optical flow tracking. Note that only frames that are successfully tracked and have a certain difference from the previous frame will be added to the sliding window. In VINS-Mono, section VI, Part D, has specific operations on marginalization of state items in the sliding window.

#### 3.1.1. Vision-Only SfM in Sliding Window

As can be seen in Figure 2, the initialization method based on the optical flow method starts on the premise that the sliding window, which serves as the image frame data buffer, has been filled with the image frames output by the camera. When the sliding window is full of frames, we check feature correspondences between the latest frame and all previous frames. If we can find stable feature tracking (more than 30 tracked features) and sufficient parallax (more than 20 pixels) between the latest frame and any other frames in the sliding window, we recover the relative rotation and up-to-scale translation between these two frames using the five-point algorithm [28]. Then, the feature points observed in the two frames are triangulated. Based on these triangulated features, a PnP method [29] is performed to estimate poses of all other frames in the window. Finally, a global full BA is carried on to minimize the total reprojection error of all feature observations. In this case, the first image frame in the sliding window is the reference coordinate system. 

#### 3.1.2. Visual-Inertial Alignment

In VINS-Mono, to scale the visual map to metric scale, the IMU preintegration is aligned with the SfM pose estimation results from the previous step, as show in Figure 3
**[8]**. The basic idea is to match the up-to-scale visual structure with IMU preintegration.

According to the spatial transformation relationship, we can easily know the coordinate transformation relationship between the camera frame and IMU frame as follows:(5)$$q_{b_{k+1}}^{b_{k}}⊗q_{c}^{b}=q_{c}^{b}⊗q_{c_{k+1}}^{c_{k}}\;=\left[Q_{1}\left(q_{b_{k+1}}^{b_{k}}\right)-Q_{2}\left(q_{c_{k+1}}^{c_{k}}\right)\right]\cdot q_{c}^{b}=Q_{k+1}^{k}\cdot q_{c}^{b}=0,$$
where
(6)$$Q_{1}\left(q\right)=\left[\begin{array}{cc} q_{w}I_{3}+{\lfloor q}_{xyz}\times \rfloor & q_{xyz}\\ -q_{xyz} & q_{w} \end{array}\right]\;Q_{2}\left(q\right)=\left[\begin{array}{cc} q_{w}I_{3}-{\lfloor q}_{xyz}\times \rfloor & q_{xyz}\\ -q_{xyz} & q_{w} \end{array}\right],$$
are matrix representation for left and right quaternion multiplication, ${\lfloor q}_{xyz}\times \rfloor$ is the skew-symmetric matrix from the first three elements $q_{xyz}$ of a quaternion, $q_{w}$ is the real part of the quaternion. Moreover, $q_{y}^{x}$ denotes the rotation quaternion from the *y* coordinate system to the *x* coordinate system, where $c_{i}$ and $b_{i}$ at positions of *x* and *y* represent the camera coordinate system and inertial coordinate system corresponding to the *i*-th frame image, respectively. 

The relation of Equation (5) holds between all adjacent frames, and we can construct the overdetermined equation according to the equality relationship between multiple frames as: (7)$$\left[\begin{array}{c} \begin{array}{c} w_{1}^{0}\cdot Q_{1}^{0}\\ w_{2}^{1}\cdot Q_{2}^{1} \end{array}\\ \begin{array}{c} \vdots \\ w_{N}^{N-1}\cdot Q_{N}^{N-1} \end{array} \end{array}\right]\cdot q_{c}^{b}=Q_{N}\cdot q_{c}^{b}=0,\;$$
where *N* is the index of the latest frame which keeps growing until the end of rotation calibration and $w_{k+1}^{k}$.is weight derived from the robust norm for a better outlier processing. Then, use singular value decomposition (SVD) to obtain the rotation relationship between IMU and camera, $q_{c}^{b}$. The extrinsic parameter results obtained by this method are not completely accurate because IMU error is not considered, and the value needs to be optimized by back-end optimization in the future. If $q_{c}^{b}$ is known exactly, skip this step. See Formula (4)–(9) in [11] for the detailed calculation process.

Gyroscope Bias Calibration

Combining the results of SfM, $q_{c}^{b}$, and IMU preintegration in Equations (2) and (4), we can construct an equation between two consecutive frames, $b_{k}$ and $b_{k+1}$:(8)$$q_{b_{k+1}}^{c_{0}}⊗q_{b_{k}}^{c_{0}}{}^{-1}=\gamma_{b_{k+1}}^{b_{k}},$$

The cost equation is constructed accordingly, while the linear equation for IMU preintegration with respect to the gyro deviation is as follows to minimize the cost function:(9)$$\underset{\delta b_{\omega }}{\min}\sum_{k\epsilon Β}∥q_{b_{k+1}}^{c_{0}}{}^{-1}⊗q_{b_{k}}^{c_{0}}⊗\gamma_{b_{k+1}}^{b_{k}}∥{}^{2},$$
where Β indexes all frames in the window. In such a way, we get an initial calibration of the gyroscope bias $b_{\omega }$. Then, we repropagate all IMU preintegration terms ${\hat{\alpha }}_{b_{k+1}}^{b_{k}}$, ${\hat{\beta }}_{b_{k+1}}^{b_{k}},$ and ${\hat{\gamma }}_{b_{k+1}}^{b_{k}}$ using the new gyroscope bias.

Velocity, Gravity Vector, and Metric Scale Initialization

We set the first camera frame ${\left(\cdot \right)}^{c_{0}}$ as the reference frame for SfM. All frame poses (${\bar{p}}_{c_{k}}^{c_{0}}$, $q_{b_{k}}^{c_{0}}$) and feature positions are represented with respect to ${\left(\cdot \right)}^{c_{0}}$. Moreover, ($p_{c}^{b}$, $q_{c}^{b}$) are extrinsic parameters between the camera and the IMU. After the gyroscope bias is initialized, the gravity vector and metric scale in the world coordinate system can be initialized by establishing the relation equation between the IMU preintegration of two consecutive frames and the camera pose information obtained by SfM as:(10)$$\begin{array}{c} \alpha_{b_{k+1}}^{b_{k}}=R_{c_{0}}^{b_{k}}\left(s\left({\bar{p}}_{b_{k+1}}^{c_{0}}-{\bar{p}}_{b_{k}}^{c_{0}}\right)+\frac{1}{2}g^{c_{0}}\Delta t_{k}^{2}-R_{b_{k}}^{c_{0}}v_{b_{k}}^{b_{k}}\Delta t_{k}\right)\\ \beta_{b_{k+1}}^{b_{k}}=R_{c_{0}}^{b_{k}}\left(R_{b_{k+1}}^{c_{0}}v_{b_{k+1}}^{b_{k+1}}+g^{c_{0}}\Delta t_{k}-R_{b_{k}}^{c_{0}}v_{b_{k}}^{b_{k}}\right) \end{array},$$
among
(11)$$\begin{array}{c} q_{b_{k}}^{c_{0}}=q_{c_{k}}^{c_{0}}⊗{\left(q_{c}^{b}\right)}^{-1}\\ s{\bar{p}}_{b_{k}}^{c_{0}}=s{\bar{p}}_{c_{k}}^{c_{0}}-R_{b_{k}}^{c_{0}}p_{c}^{b} \end{array}$$

According to the above equation, and the velocity $v_{b_{i}}^{b_{i}}$, gravity vector $g^{c_{0}}$ and scale s are taken as the state quantities to be estimated as:(12)$$\chi_{I}=\left[v_{b_{0}}^{b_{0}},v_{b_{1}}^{b_{1}},\;\;\;v_{b_{n}}^{b_{n}},g^{c_{0}},s\right],$$

Combine Equations (10) and (11) into the following linear measurement model:(13)$${\hat{z}}_{b_{k+1}}^{b_{k}}=\left[\begin{array}{c} {\hat{\alpha }}_{b_{k+1}}^{b_{k}}-p_{c}^{b}+R_{c_{0}}^{b_{k}}R_{b_{k+1}}^{c_{0}}p_{c}^{b}\\ {\hat{\beta }}_{b_{k+1}}^{b_{k}} \end{array}\right]=H_{b_{k+1}}^{b_{k}}\chi_{I}+n_{b_{k+1}}^{b_{k}},$$
where
(14)$$H_{b_{k+1}}^{b_{k}}=\left[\begin{array}{ccc} -I\Delta t_{k}\;\;\;0 & \frac{1}{2}R_{c_{0}}^{b_{k}}\Delta t_{k}^{2} & R_{c_{0}}^{b_{k}}\left({\bar{p}}_{b_{k+1}}^{c_{0}}-{\bar{p}}_{b_{k}}^{c_{0}}\right)\\ -I\;\;\;R_{c_{0}}^{b_{k}}R_{b_{k+1}}^{c_{0}} & & \begin{array}{cc} R_{c_{0}}^{b_{k}}\Delta t_{k} & 0 \end{array} \end{array}\right],$$

$R_{b_{k}}^{c_{0}}$, $R_{b_{k+1}}^{c_{0}}$, ${\bar{p}}_{c_{k},}^{c_{0}}$ and ${\bar{p}}_{c_{k+1}}^{c_{0}}$ are obtained from the up-to-scale monocular visual SfM, and $\Delta t_{k}$ is the time interval between two consecutive frames. By solving this linear least-square problem:(15)$$\underset{\chi_{I}}{\min}\sum_{k\epsilon Β}∥{\hat{z}}_{b_{k+1}}^{b_{k}}-H_{b_{k+1}}^{b_{k}}\chi_{I}∥{}^{2},$$
the body frame velocities for every frame in the window, the gravity vector in the visual reference frame ${\left(\cdot \right)}^{c_{0}}$, as well as the scale parameter can be obtained.

Gravity Refinement

Then, the gravity is perturbed with two variables on its tangent space, which preserves 2-DOF as in Equation (16). The aim is to optimize the direction of the gravity vector obtained in the initialization of the previous step:(16)$$g\left(\bar{\hat{g}}+\delta g\right),\delta g=\omega_{1}b_{1}+\omega_{2}b_{2},$$
where g is the known magnitude of the gravity, and $\bar{\hat{g}}$ is the unit vector representing the direction of gravity. Moreover, $b_{1}$ and $b_{2}$ are two orthogonal bases spanning the tangent plane, as shown in Figure 4 [8], and $\omega_{1}$ and $\omega_{2}$ are 2-D perturbation toward $b_{1}$ and $b_{2}$, respectively. Lastly, $b_{1}$ and $b_{2}$ can be set arbitrarily as long as they are not parallel. 

Then, the g in Equation (10) is replaced by $g\left(\bar{\hat{g}}+\delta g\right)$, and solve for 2-D δg together with other state variables. This process iterates several times until $\hat{g}$ converges.

Completing Initialization

The rotation $q_{c_{0}}^{w}$ between the world frame and the camera frame $c_{0}$ can be obtained by rotating the refined gravity vector $g^{c_{0}}$ to the z-axis. Then, all variables from the reference frame ${\left(\cdot \right)}^{c_{0}}$ are rotated to the world frame ${\left(\cdot \right)}^{w}$. The body frame velocities will also be rotated to the world frame. Translational components from the visual SfM will be scaled to metric units. At this point, the initialization procedure is finished.

### 3.2. Initialization Framework for Monocular VO in ORB Feature-Based Method

After the detection and description of feature points in an image frame is finished, feature matching is carried out. 

In this paper, the ORB initialization framework only carries out the pure vision initialization process as in ORB-SLAM2.

The monocular initialization process of ORB-SLAM2 can be divided into the following three stages.

#### 3.2.1. Select two Frames as Initial Reference Frames

The first reference frame selected during initialization requires a sufficient number of feature points detected (>100); there is also a requirement on the number of feature point detection for the following image frames. If it does not meet the requirement on number of feature points in the two consecutive frames, the first reference frame should be selected again, and the above process should be repeated.

#### 3.2.2. Calculation of Relative Pose Based on the Matching Features and Triangulation

When the number of feature points in two consecutive frames is sufficient, DBOW2 bag-of-word is used to accelerate the matching process of feature points. After word bag matching, multi-thread parallel processing is used to calculate the relative pose using basic matrix model and homography matrix model, respectively, and the pose transformation results of the two frames are obtained. The reprojection error of features is used to score the two pose results. The pose results obtained by the model with better score are selected to complete the triangulation of matching points. Then, the PnP algorithm is used to calculate the relative poses of the new image. 

#### 3.2.3. Global BA

Global BA is performed on the previously recovered feature points and poses to optimize the initialized map. 

So far, we can obtain an initial up-to-scale map based on the ORB feature-based method that saves the relative poses and 3D coordinates of the corresponding feature points between image frames.

From the above description, we can see that the initialization process of the visual map based on ORB feature points is simple and fast, but the established map is not in metric units and does not align with the inertial coordinate system. On the premise that the two initialization processes are carried out synchronously, the map initialization based on ORB feature points should be completed first.

### 3.3. The Initialization Framework That Fuses the Initial Results of the Optical Flow Method and Feature Point Methods 

The data fusion operation proposed in this paper will be performed after the initialization of both the optical flow method and the feature-based method. Since the initialization of the feature-based method only depends on the image of the monocular camera, the visual map established by feature-based method needs to be multiplied by a fixed scale to be in metric units. This map is referred to as the feature-based map in the following. So, the map established by optical flow method is referred to as the optical flow map. 

The fusion process can be divided into three steps, as shown in Figure 5. First, the similarity matrix transformation from feature-based map is obtained by the Umeyama algorithm. The Umeyama algorithm is designed to compute the positional relationship between two sets of data. When aligning two trajectories $x_{i}$ and $y_{i}$, the goal is to compute a set of s, R, and t such that the objective function (17) is optimal:(17)$$\frac{1}{n}\sum_{i=1}^{n}∥x_{i}-{\left(sRy_{i}+t\right)∥}_{2}^{2},$$
where s is the scaling factor, R is the rotation matrix, and t represents the translation. 

Then, the transfer matrix between two co-view frames in the feature-based map is substituted into the corresponding relationship in optical flow method for optimization. Finally, the pose and rotation information of the state vector optimized in the previous step is fixed, and the remaining states in the state vector are further optimized.

#### 3.3.1. Rough Map Alignment Using Umeyama Algorithm

In order to better perform the data fusion operation between the two maps, we use the Umeyama algorithm to solve the similarity transformation matrix, which is solved by the inputs of position information from the common key frame in the two maps. The basic idea of this step is to restore the feature-based map to metric unit and align with the world coordinate system in optical flow map through scaling, rotation, and translation operations.

Through the Umeyama algorithm, we can get the similarity transformation matrix $\left[\begin{array}{cc} sR_{w_{orb}}^{w_{vins}} & t\\ 0 & 1 \end{array}\right]$ and make the following equation between the two maps approximately true:(18)$$\begin{array}{c} {\hat{P}}_{c_{k}}^{w_{vins}}\approx sR_{w_{orb}}^{w_{vins}}P_{c_{k}}^{w_{orb}}+t\\ {\hat{R}}_{c_{k}}^{w_{vins}}\approx R_{w_{orb}}^{w_{vins}}R_{c_{k}}^{w_{orb}} \end{array},$$

The feature-based map can be approximate aligned with the optical flow map after the similarity transformation above, and the alignment process is shown in Figure 6.

#### 3.3.2. Local BA in Visual States

The feature-based map mainly saves the relative rotation and translation information between frames. Due to the higher accuracy of the feature matching algorithm used in the initialization of feature-based method, the optimization of the states in feature-based map is ignored. The relevant information provided by the feature-based map is added to the optimization as measurement.

Thus, the state vector in the local BA is defined as:
(19)$$\begin{array}{c} \chi_{L}=\left[y_{0},y_{1},\ldots y_{n},\lambda_{0,}\ldots \lambda_{m},x_{c}^{b},s\;\;\;R_{w_{ins}}^{w_{orb}}\right]\\ y_{k}=\left[p_{b_{k}}^{w_{ins}},q_{b_{k}}^{w_{ins}}\right]\\ x_{c}^{b}=\left[p_{c}^{b},q_{c}^{b}\right] \end{array}$$
where $y_{k}$ is the position state of IMU states at the time the *k*th image is captured. It contains position and orientation of the IMU in the world frame. Moreover, s is scale value in the similarity matrix which calculated by Umeyama algorithm and needs to be optimized in BA, *n* is the total number of keyframes, and *m* is the total number of features tracked by optical flow method in the sliding window. Lastly, $\lambda_{l}$ is the inverse distance of the *l*th features from its first observation.

The visual BA formulation can be optimized by minimizing the sum of the Mahalanobis norm of all measurement residuals to obtain a maximum posteriori estimation as: (20)$$\begin{array}{r} \underset{\chi }{\min}\left\{\sum_{\left(i,j\right)\in \psi }\left(∥r_{\psi }{\left({\hat{q}}_{c_{j}}^{c_{i}},\chi_{L}\right)∥}^{2}+∥r_{\psi }{\left(\left(P_{c_{j}}^{w_{orb}}-P_{c_{i}}^{w_{orb}}\right),\chi_{L}\right)∥}^{2}\right)\right.\\ \left.+\sum_{\left(l,j\right)\in C}\rho \left(∥r_{C}{\left({\hat{z}}_{l}^{c_{j}},\chi_{L}\right)∥}_{P_{l}^{c_{j}}}^{2}\right)\right\} \end{array},$$
where the Huber norm is defined as:(21)$$\rho \left(e\right)=\left\{\begin{array}{cc} \begin{array}{c} e \end{array} & e\leq 1\\ 2\sqrt{e}-1 & e>1 \end{array}\right.,$$
where $r_{\psi }\left(\left(P_{c_{j}}^{w_{orb}}-P_{c_{i}}^{w_{orb}}\right),\chi_{L}\right)$, $r_{\psi }\left({\hat{q}}_{c_{j}}^{c_{i}},\chi_{L}\right),$ and $r_{C}\left({\hat{z}}_{l}^{c_{j}},\chi_{L}\right)$ are residuals for scale estimation in feature-based map, rotation estimation between two common-view frames in feature-based map, and visual measurements in optical flow map, respectively. The detailed definition of residual term will be presented next.

Residuals for Rotation Estimation between Two Common-view Frames in feature-based Map

The position parameters of the image frames in the optical flow map are mainly optimized based on the relative rotation between frames solved by the feature-based map. According to the inter-frame conversion matrix of the feature-based map, combined with Equation (18), the constrained equation holds as follow:(22)$$\begin{array}{c} r_{\psi }\left({\hat{q}}_{c_{j}}^{c_{i}},\chi \right)={\left[q_{c_{j}}^{w_{vins}}{}^{-1}⊗q_{c_{i}}^{w_{vins}}⊗\left({\hat{q}}_{c_{j}}^{c_{i}}\right)\right]}_{xyz}\\ ={\left[{\left(q_{b_{j}}^{w_{vins}}⊗q_{c}^{b}\right)}^{-1}⊗\left(q_{b_{i}}^{w_{vins}}⊗q_{c}^{b}\right)⊗\left({\hat{q}}_{c_{j}}^{w_{vins}}⊗{\hat{q}}_{c_{j}}^{w_{vins}}{}^{-1}\right)\right]}_{xyz}\\ ={\left[q_{c}^{b}{}^{-1}⊗q_{b_{j}}^{w_{vins}}{}^{-1}⊗q_{b_{i}}^{w_{vins}}⊗q_{c}^{b}⊗\left(q_{w_{orb}}^{w_{vins}}⊗q_{c_{j}}^{w_{orb}}⊗q_{c_{i}}^{w_{orb}}{}^{-1}⊗q_{w_{orb}}^{w_{vins}}{}^{-1}\right)\right]}_{xyz}, \end{array}$$
where ${\hat{q}}_{c_{j}}^{w_{vins}}$, $q_{w_{orb}}^{w_{vins}},$ and $q_{c_{j}}^{w_{orb}}$ are the quaternion form of ${\hat{R}}_{c_{k}}^{w_{vins}}$, $R_{w_{orb}}^{w_{vins}},$ and $R_{c_{j}}^{w_{orb}}$, respectively. ${\hat{R}}_{c_{k}}^{w_{vins}}$ is calculated in Equation (18). The *i*th image is co-viewed with the *j*th image. Lastly, ψ is the set of keyframe pairs in the feature-based map with co-view feature points.

Residuals for Scale Estimation in ORB Map

Since the accuracy of the feature-based map on inter-frame rotation matrix is considerable, the translation requires a scale factor to restore it. 

According to the feature-based map and the similarity matrix provided by Umeyama algorithm in the first step, the translation between two co-view frames and scale factor s are added to the residual formulation. Combined with Equation (18), the residuals corresponding to s are defined as: (23)$$r_{s}\left(\left(P_{c_{j}}^{w_{orb}}-P_{c_{i}}^{w_{orb}}\right),\chi_{L}\right)=\frac{P_{c_{j}}^{w_{vins}}-P_{c_{i}}^{w_{vins}}}{{\hat{P}}_{c_{j}}^{w_{vins}}-{\hat{P}}_{c_{i}}^{w_{vins}}}-1\;=\frac{\left|P_{b_{j}}^{w_{vins}}+R_{b_{j}}^{w_{vins}}p_{c}^{b}-P_{b_{i}}^{w_{vins}}-R_{b_{i}}^{w_{vins}}p_{c}^{b}\right|}{\left|s*\left(P_{c_{j}}^{w_{orb}}-P_{c_{i}}^{w_{orb}}\right)\right|}-1,$$
where $R_{b_{j}}^{w_{vins}}$ is the transformation matrix from the inertial coordinate system of the *j*th image frame to the world coordinate system of the optical flow map. The reason for considering the inertial coordinate system is to further optimize the extrinsic parameters between the camera and the IMU. Lastly, $\left|\cdot \right|$ denotes the norm of the vector.

Visual Measurement Residual for features tracked by Optical flow

Consider the *l*th feature that is first observed in the *i*th image, the residual for the feature observation in the *j*th image is defined as:(24)$$r_{C}\left({\hat{z}}_{l}^{c_{j}},\chi \right)=P_{l}^{c_{j}}-{\hat{P}}_{l}^{c_{j}}\;{\hat{P}}_{l}^{c_{j}}=\pi_{c}{}^{-1}\left(\left[\begin{array}{c} {\hat{u}}_{l}^{c_{j}}\\ {\hat{v}}_{l}^{c_{j}} \end{array}\right]\right)\;P_{l}^{c_{j}}=R_{b}^{c}\left(R_{w}^{b_{j}}\left(R_{b_{i}}^{w}\left(R_{c}^{b}\frac{1}{\lambda_{l}}\pi_{c}{}^{-1}\left(\left[\begin{array}{c} {\hat{u}}_{l}^{c_{i}}\\ {\hat{v}}_{l}^{c_{i}} \end{array}\right]\right)+p_{c}^{b}\right)+p_{b_{i}}^{w}-p_{b_{j}}^{w}\right)-p_{c}^{b}\right)$$
where $\left[\begin{array}{cc} \begin{array}{c} {\hat{u}}_{l}^{c_{i}} \end{array} & {\hat{v}}_{l}^{c_{i}} \end{array}\right]$ is the first observation of the *l*th feature that happens in the *i*th image, $\left[\begin{array}{cc} \begin{array}{c} {\hat{u}}_{l}^{c_{j}} \end{array} & {\hat{v}}_{l}^{c_{j}} \end{array}\right]$ is the observation of the same feature in the *j*th image, $\pi_{c}{}^{-1}$ is the back projection function, which turns a pixel location into a unit vector using camera intrinsic parameters, ${\hat{\bar{P}}}_{l}^{c_{j}}$ is the unit vector for the observation of the *l*th feature in the *j*th frame, and $P_{l}^{c_{j}}$ is predicted feature measurement on the unit sphere by transforming its first observation in the *i*th frame to the *j*th frame.

#### 3.3.3. Full BA with Some Optimized States Fixed

In the previous step, the inter-frame constraint in the feature-based map is introduced to optimize the state of image frame and extrinsic parameters in optical flow map. The optimized states will be fixed, and then the Full BA operation of visual states in optical flow map and inertial states is performed:(25)$$\begin{array}{l} \chi_{F}=\left[\begin{array}{cc} \begin{array}{cc} \begin{array}{cc} {\text{x}}_{0}, & {\text{x}}_{1} \end{array}, & \begin{array}{cc} \ldots & \begin{array}{ccc} {\text{x}}_{n}, & \begin{array}{cc} \lambda_{0,} & \ldots \end{array} & \lambda_{m} \end{array} \end{array} \end{array}, & \begin{array}{c} \begin{array}{c} {\text{x}}_{c}^{b} \end{array} \end{array} \end{array}\right]\\ {\text{x}}_{k}=\left[\begin{array}{cc} \begin{array}{ccc} p_{b_{k}}^{w} & v_{b_{k}}^{w} & q_{b_{k}}^{w} \end{array} & \begin{array}{cc} b_{a} & b_{g} \end{array} \end{array}\right]\\ \begin{array}{c} \begin{array}{c} {\text{x}}_{c}^{b}=\left[\begin{array}{cc} p_{c}^{b} & q_{c}^{b} \end{array}\right] \end{array} \end{array} \end{array}$$
where $x_{k}$ is the IMU state at the time that the *k*th image is captured. It contains position, velocity, and orientation of the IMU in the world frame, and the acceleration bias and gyroscope bias in the IMU body frame.

The visual measurement model here is same as in Local BA, as Equation (24). The only difference is that the position information ($q_{b}^{w_{vins}},P_{b}^{w_{vins}}$) of the frame that exits in both the optical flow map and the feature-based map is treated as a constant. To this end, the nonlinear cost function here is formulated as: 

(26)$$\begin{array}{c} \underset{\chi }{\min}\left\{\underset{\left(l,j\right)\in C}{\sum }\rho \left(∥r_{C}{\left({\hat{z}}_{l}^{c_{j}},\chi_{F}\right)∥}_{P_{l}^{c_{j}}}^{2}\right)+\underset{k\in B}{\sum }∥r_{B}{\left({\hat{z}}_{b_{k+1}}^{b_{k}},\chi_{F}\right)∥}_{P_{b_{k+1}}^{b_{k}}}^{2}\right\} \end{array},$$
where $r_{B}\left({\hat{z}}_{b_{k+1}}^{b_{k}},\chi_{F}\right)$ and $r_{C}\left({\hat{z}}_{l}^{c_{j}},\chi_{F}\right)$ are residuals for IMU and visual measurements, respectively. Moreover, $r_{C}\left({\hat{z}}_{l}^{c_{j}},\chi_{F}\right)$ is defined in (24), $r_{B}\left({\hat{z}}_{b_{k+1}}^{b_{k}},\chi_{F}\right)$ is defined as: (27)$$r_{B}\left({\hat{z}}_{\begin{array}{c} b_{k+1} \end{array}}^{b_{k}},\chi_{F}\right)=\left[\begin{array}{c} \delta \alpha_{b_{k+1}}^{b_{k}}\\ \delta \beta_{b_{k+1}}^{b_{k}}\\ \delta \theta_{b_{k+1}}^{b_{k}}\\ \delta b_{a}\\ \delta b_{g} \end{array}\right]\;=\left[\begin{array}{c} R_{w}^{b_{k}}\left(p_{b_{k+1}}^{w}-p_{b_{k}}^{w}+\frac{1}{2}g^{w}\Delta t_{k}^{2}-v_{b_{k}}^{w}\Delta t_{k}\right)-{\hat{\alpha }}_{b_{k+1}}^{b_{k}}\\ R_{w}^{b_{k}}\left(v_{b_{k+1}}^{w}+g^{w}\Delta t_{k}-v_{b_{k}}^{w}\right)-{\hat{\beta }}_{b_{k+1}}^{b_{k}}\\ 2{\left[q_{b_{k}}^{w}{}^{-1}⊗q_{b_{k+1}}^{w}⊗{\left({\hat{\gamma }}_{b_{k+1}}^{b_{k}}\right)}^{-1}\right]}_{xyz}\\ b_{a}{}_{b_{k+1}}-b_{a}{}_{b_{k}}\\ b_{\omega }{}_{b_{k+1}}-b_{\omega }{}_{b_{k}} \end{array}\right]$$
where ${\left[\cdot \right]}_{xyz}$ extracts the vector part of a quaternion q for the error-state representation, $\delta \theta_{b_{k+1}}^{b_{k}}$ is the 3-D error-state representation of quaternion, and $\left[\begin{array}{ccc} {\hat{\alpha }}_{b_{k+1}}^{b_{k}}, & {\hat{\beta }}_{b_{k+1}}^{b_{k}},\; & {\hat{\gamma }}_{b_{k+1}}^{b_{k}} \end{array}\right]$ are pre-integrated IMU measurement terms between two consecutive image frames. Accelerometer and gyroscope biases are also included in the residual terms for the online correction.

## 4. Experiments and Discussion

To evaluate the effectiveness and practicability of the improved initialization of monocular visual-inertial SLAM, this paper conducts experiments on the visual-inertial datasets (EuRoC) collected on-board a micro aerial vehicle (MAV). The datasets contain stereo images, synchronized IMU measurements, and accurate motion and structure ground-truth [30]. The main experiments are carried out in three indoor sequences of EuRoC dataset which were recorded with a micro aerial vehicle equipped with two global-shutter, monochrome cameras, and an IMU. The two cameras and the IMU were hardware time-synchronized and were logged at a rate of 20 Hz and 200 Hz, respectively. 

The sequences we chose contain the following three cases in the initialization phase: (1) MH_01_easy has a fast translational motion in the initialization phase; (2) MH_04_difficult starts from static in the initialization stage, and the weak texture part accounts for a large proportion in the starting screen; (3) The initialization phase of V1_03_difficult starts with a stationary phase followed by a fast, large rotation motion. The above three scenarios are enough to cover most application scenarios. All the experiments are carried out with an Intel CPU i7-8565U (8 cores @1.8GHz) laptop computer with 16GB RAM.

### 4.1. Implementation Details

The proposed initialization method is implemented in the combination of VINS-Mono and ORB-SLAM2. The VINS-Mono is deeply combined with the monocular mode of ORB-SLAM2 to realize the synchronous input and processing of data. Then, the initialization process based on the optical flow method in VINS-Mono and the feature based method in ORB-SLAM2 can be carried out synchronously. 

The specific fusion optimization algorithm is shown at Section 3. Firstly, the Umeyama algorithm is used to estimate the scale of ORB-SLAM2 based on the matching trajectories obtained by the two initialization threads. Then, the constraints provided by the trajectory which is obtained from the feature-based method after scaling to metric unit are added to the modified back-end optimization of VINS-Mono. Finally, the optimized state estimates are obtained after several iterations.

### 4.2. Validation of Umeyama Algorithm

In order to verify the effectiveness of the Umeyama algorithm, five tries were performed in the MH_01_easy, MH_04_difficult, and V1_03_difficult sequences, respectively. The comparison between the Umeyama estimated scale values and the actual scale values compared with trajectory truth are shown in Table 1. The scale is determined by the choice of two initialization reference frames, which is different in each time of initialization phase in ORB-SLAM2. The scale to scaling the initialization trajectory of ORB-SLAM2 to metric unit is not the same for each experiment. The ratio in Table 1 represents the scale error between the trajectory of ORB-SLAM2 scaled by the value estimated by Umeyama and the trajectory truth in percentage. It is calculated as Equation (28).
(28)$$S=\frac{S_{truth}}{S_{Umeyama}},$$

From the value of ratio, we can see that the scale estimated by Umeyama is relatively accurate compared with the actual value. To further verify, we compare the ratio value in Table 1 with the ratio value of the ORB-SLAM3 monocular inertial mode trajectory. The ratio value of ORB-SLAM3 is obtained by EVO (Python package for the evaluation of odometry and SLAM) [31]. As shown in Figure 7, the accuracy of the estimated scale is similar to ORB-SLAM3 or even better than ORB-SLAM3. ORB-SLAM3 as the state-of-the-art SLAM system is enough to prove the effect of Umeyama algorithm. 

### 4.3. Evaluation on Optimization Results

The evaluation method is carried out by EVO_APE of EVO, which is mainly used to evaluate the global consistency of the whole trajectory. APE stands for absolute pose error. By default, it executes ATE (absolute trajectory error). To evaluate this error results, we mainly compare from five aspects: max (maximum error), mean (mean value of error), RMSE (root mean square error), SSE (sum of squares for error), and STD (the standard deviation). Each index uses the mean of the five experiments, and the comparison results are shown in Figure 8; (a), (b), and (c) are the comparison results in sequence MH_01_easy, MH_04_difficult, and V1_03_difficult, respectively. It can be found that the optimized trajectory obtained by our proposed method has comparable accuracy with the trajectory obtained by ORB-SLAM3 initialization, while the trajectory obtained by VINS-Mono initialization has poor accuracy. Because VINS-Mono uses the method that the initialization phase completes the coarse estimation of the IMU bias and extrinsic parameters, and then continues to modify these states in the subsequent back-end optimization process, this method cannot guarantee a good positioning accuracy in the initialization phase. In addition, due to the design of parallel threads, an advantage of our proposed method is that even when the extrinsic parameters between the camera and the IMU are unknown, the map construction and tracking process can be started by ORB-SLAM2 in monocular mode, and the scale scaling operation will be realized after the data fusion optimization with the states estimate by initialization process of VINS-Mono.

In order to show the trajectory comparison of the proposed method more intuitively, we take V1_03_difficult Sequence as an example. The trajectory comparison diagram is shown in Figure 9; (a) is the display of the trajectory in 3D space, and (b) is the error comparison in the XYZ axis. As can be seen from Figure 9, our proposed method is comparable to the monocular-inertial mode of ORB-SLAM3. There are some discontinuities in the trajectory because we only use inertial data in the initialization stage, and the combination algorithm with optical flow method and inertia has not been implemented in the subsequent tracking stage. Therefore, the subsequent trajectory is actually the trajectory after ORB-SLAM2 was scaled to metric unit, and its tracking process will be lost due to insufficient number of feature points, but it can continue to locate after relocation. The discontinuity of VINS-Mono is because the saved frame pose data is saved together with the key frames in ORB-SLAM2, so the discontinuity also occurs. 

## 5. Conclusions

In this paper, an improved initialization method for monocular visual-inertial SLAM is proposed. It uses two visual processing methods to initialize in parallel, which improves the robustness of system initialization and the utilization of image information. The extrinsic parameter estimation algorithm in VINS-Mono is used to improve the portability of SLAM. The initialization accuracy of the proposed method is comparable to that of ORB-SLAM3. The experimental results show that the proposed method features the same scale estimation accuracy and positioning performance as ORB-SLAM3 in monocular inertia model. In addition, the extrinsic parameter estimation of the camera and the IMU is added, which enhances the portability of the proposed method. It is planned to further integrate the optical flow information, feature point information, and inertia information in the subsequent tracking stage to achieve robust and lightweight monocular visual inertia SLAM.

## Figures and Tables

**Figure 1 sensors-22-08307-f001:**
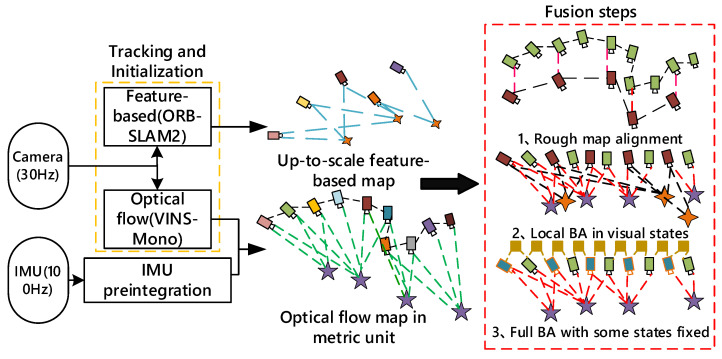
The framework of the robust parallel initialization methods.

**Figure 2 sensors-22-08307-f002:**
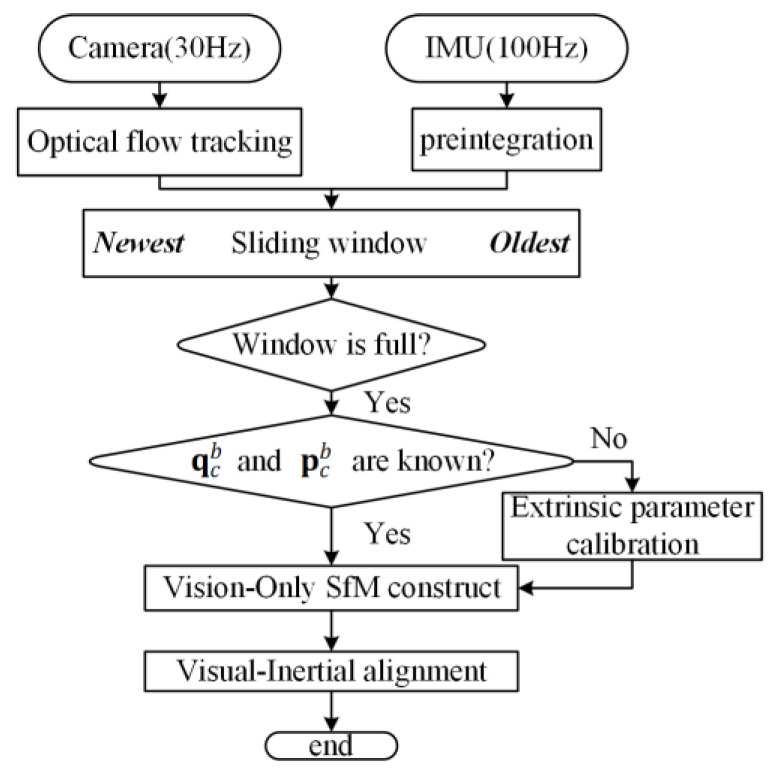
The initialization framework of VINS-Mono, including extrinsic calibration.

**Figure 3 sensors-22-08307-f003:**
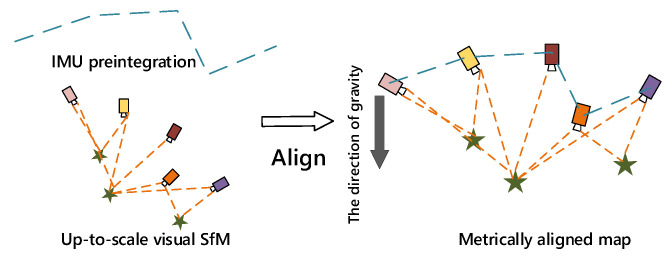
Illustration of the visual-inertial alignment process for initialization.

**Figure 4 sensors-22-08307-f004:**
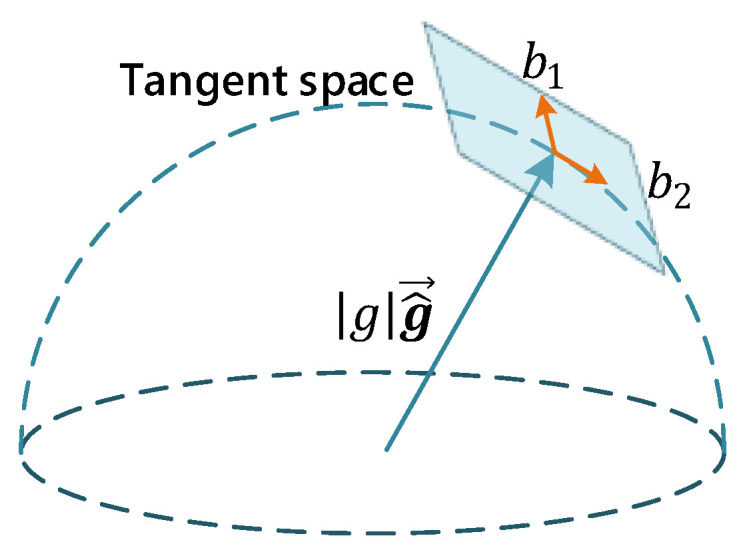
Illustration of 2-DOF perturbation of gravity. The magnitude of gravity is known.

**Figure 5 sensors-22-08307-f005:**
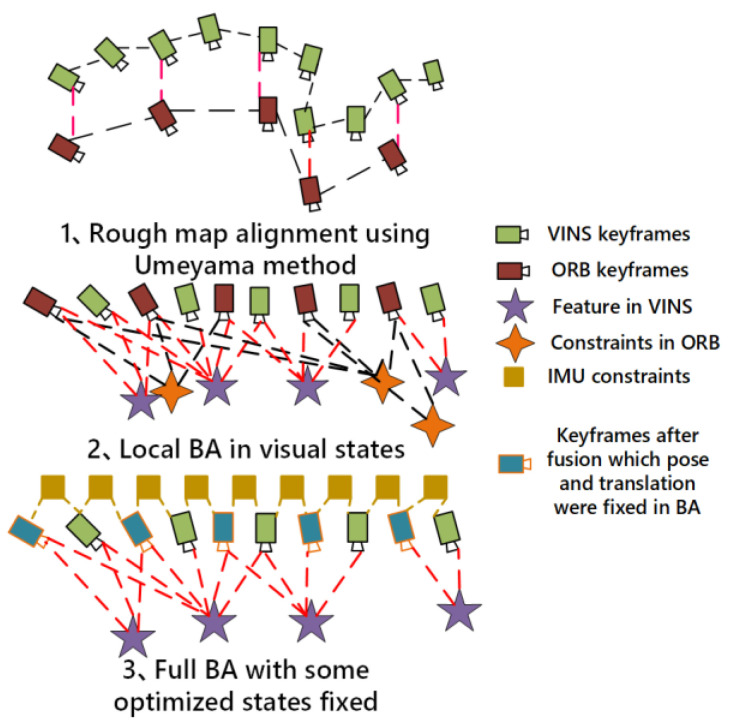
The illustration of data fusion process between the feature-based map and the optical map.

**Figure 6 sensors-22-08307-f006:**
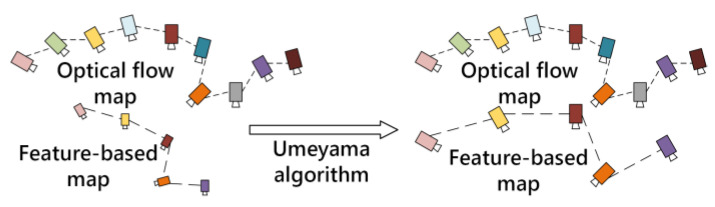
Illustration of the alignment process between optical flow map and feature-based map by using Umeyama algorithm.

**Figure 7 sensors-22-08307-f007:**
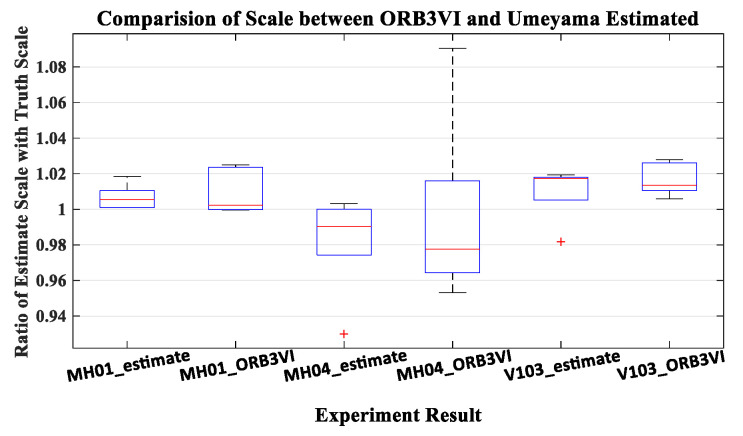
The distribution of ratio value in five times experiments. The experiments are carried out in MH_01_easy, MH_04_difficult, and V1_03_difficult sequences. The XX_estimate represents the ratio estimated by Umeyama algorithm, and the XX_ORB3VI represents the ratio obtained by ORB-SLAM3 in monocular-inertial mode.

**Figure 8 sensors-22-08307-f008:**
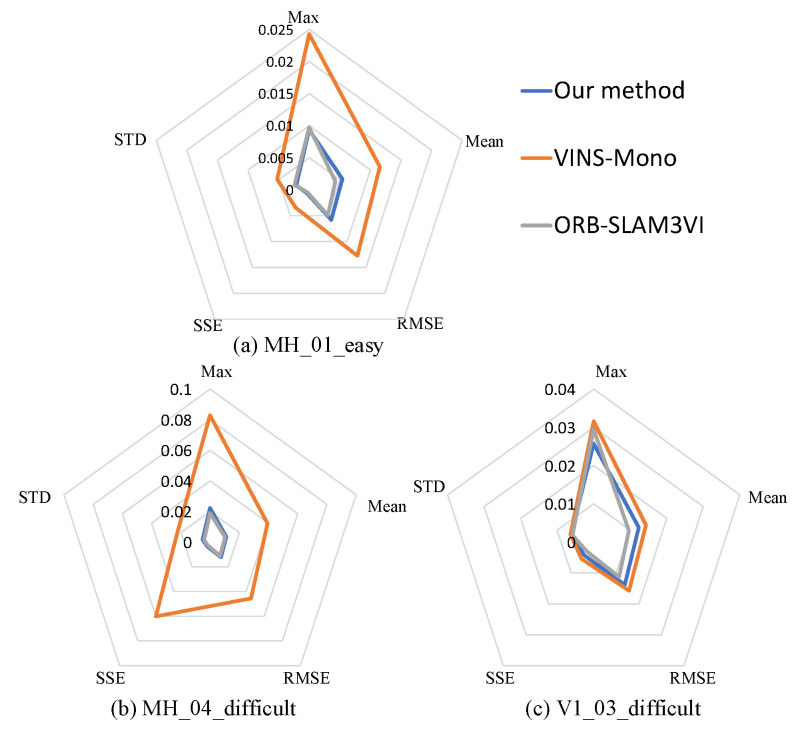
The comparison of trajectories between our method, VINS-Mono, and ORB-SLAM3 with the trajectory truth in absolute trajectory error, respectively.

**Figure 9 sensors-22-08307-f009:**
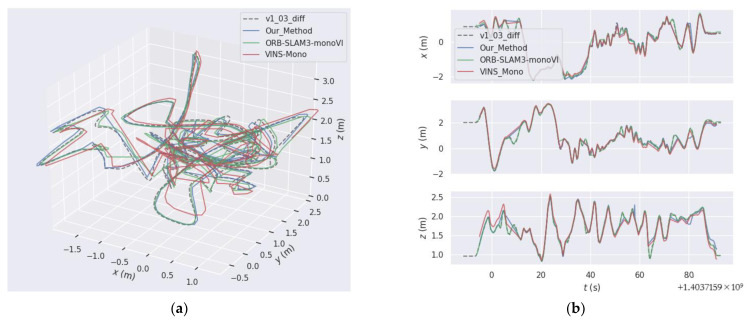
The comparison of trajectories between our method, VINS-Mono and ORB-SLAM3 with the trajectory truth, v1_03_diff. (**a**) trajectory absolute errors shown in 3D space; (**b**) trajectory absolute errors shown in x, y and z axes, respectively.

**Table 1 sensors-22-08307-t001:** Scale Comparision of Umeyama Estimated against the Truth.

Sequence	Times	$S_{Umeyama}$	$S_{truth}$	S
MH01_easy	1	0.7238	0.7246	1.0011
	2	4.0751	4.1501	**1.0184**
	3	4.0546	4.0581	1.0009
	4	6.2273	6.2763	1.0079
	5	3.8913	3.9124	1.0054
MH04_difficult	1	9.5073	9.4969	0.9989
	2	11.046	10.2725	**0.9300**
	3	8.6863	8.7148	1.0033
	4	10.0800	9.98360	0.9904
	5	9.5568	9.45228	0.9891
V103_difficult	1	2.0469	2.08666	**1.0194**
	2	3.8243	3.8904	1.0173
	3	2.2503	2.2093	0.9818
	4	2.0506	2.0868	1.0177
	5	2.8825	2.9198	1.0129

## Data Availability

Not applicable.

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
