# Peer review of "A Robust Parallel Initialization Method for Monocular Visual-Inertial SLAM"

_sensors, 2022, doi:10.3390/s22218307_

Round 1
Reviewer 1 Report
The paper the usage of the measurements of different types in the SLAM. Namely, the authors consider the measurements obtained by the inertia sensor and the gyroscope and visual information perceived by on-board camera. The obtained combination of the data obtained from different sensors is used for initialization of the SLAM algorithms.
The authors claim that the suggested techniques result in more accurate and robust initialization of the SLAM and so – in more effective positioning of the robots.
In my opinion, the theme of the paper and the methods are very interesting and important, and if the authors are right, they can lead to serious progress in the studies of robots positioning in uncertain environment.
However, since the paper is written in very unclear manner, I cannot check the presented reasoning and cannot guarantee that the results are correct.
In particular, the description of the preprocessing algorithms in section 2.1 is too brief and does not include all details of the methods. In my opinion, after indicating the methods it will be useful to outline the algorithms used for tracking optical flow (section 2.1.1) and for detecting and matching (section 2.1.2). Certainly, the outlines may be presented in the Appendices.
In section 2.2, the preprocessing methods are presented in the form of commutable formulas, but some of the symbols are not defined. In the equations (1) symbols a, R, g and omega (all without hat) are not defined and even not clarified. Also, the reasoning of such definition is not clear: on one hand, it is rather close to regression equations or more generally – to Kalman filter, but on the other hand, each equation includes two components with Gaussian noise with known parameters that allows simple filtering. Equations (2)-(4) are also very unclear and, in my opinion, require detailed explanations both of the formulas and of the underlying reasoning. The reference in the form “refer to VINS-Mono for specific 155 definitions” (lines 155-156) is, certainly, not enough for understanding the meaning of the equations and authors’ ideas.
Section 3 includes rather unclear phrases like “When sliding window is full of frames, we check feature correspondences between 173 the latest frame and all previous frames” (lines 173-174) – what does it mean “is full of frames”? How you check? What is the measure of feature correspondence? And rather unclear equations – what is q? and what do you mean by operation in eq. (5) and below? What is the “principle of robot hand-eye calibration”? Reference to the paper [11] is not enough; in my opinion, the principle must be presented in the paper. Starting from eq. (8) I cannot follow the reasoning in this section since most of the symbols are undefined.
What is “Umeyama algorithm” appearing in section 3?
Validation and discussion (section 4) sound reasonable, but, as above, the procedures should be outlined more clearly.
Summarizing, the paper requires major revisions and corrections of the form of presentation. After corrections that will make presentation clear and understandable, the paper can be considered again, and the presented methods and results should be checked.
Reviewer 2 Report
Extensive work is still required for the manuscript. Particularly author need to introduce all variables, for example those in (2). In addition, more evaluation from figure is required. Instead of only present “as shown in Figure XX”. What kind of information author does expect reader could know from figure? What is the step-by-step procedure?
One key question, author mention that the work improves VINS-Mono algorithm. The original model of VINS-Mono shall present (as separated subsection) in the manuscript, and author should state what are the changes have been done for the improvement.
1. Acronyms are needed to be properly defined when they are firstly introduced. For example, what is VINS stand for? What is ORB?
2. In line 91, author shall provide more details on “overall framework is too large”. Is it in term of computation, memory, speed, stack size or what? And also, what does resource constraint platform mean? As there are many possible types of resources could be constrained.
3. In introduction, author should mention which other algorithm will be compared, what are the key criteria that will be investigated during the comparison process. If there is no other algorithm being compared, author shall consider different algorithm to compare and to benchmark his/her algorithm
4. In line 133, author should specific the source of these methods.
5. What are the primary variables or items that author want to initialize?
6. What is the subscript “t” in equation (1)?
7. Author needs to shows the relationship between (2) and (4), or provide appropriate references.
8. Reference is needed for figure if it is not originally generated by author
9. Section 4.2 could causes confuse to reader as author suddenly evaluating the effectiveness of umeyama algorithm. But in fact, improving the initialization process is supposedly the key component of this manuscript.
10. Is ORB-SLAM3 data considered as truth in Table 3?
11. Author shall explain what are the differences between each dataset (or what kind of configuration environment being used).
Round 2
Reviewer 1 Report
Thank you for the efforts, in my opinion in the current form the paper sounds better and can be understood in more details.
Together with that in my opinion the variables appearing in the equations (1)-(4) must be clarified additionally. In equation (4) appears quaternion multiplication, thus the variable appearing in the equation are quaternions. What are the elements of these quaternions?
By the way, quaternion multiplication used in equation (4) is defined only after equations (5)-(6).
Equation (29) sounds strange. Can you clarify it please.
The are some minor misprints like
L160-161 – missed “to”: The method is to calculate the matching relationship of ORB feature points extracted between two frames, and then [to] establish the initialized…
L202 – missed “basing” (or “following” instead of “on”): Equation (2) is derived [basing] on the assumption…
L232: - missed “n”: is show[n] in Figure 2
L240: “… a sliding window size of 10 …” Do you mean ten pixels?
L278: “Where” should be “where”
L279: “calibration.” should be “calibration,” (comma instead of point); “.is” should be “ is” (without point before “is”)
L303: “Conbine” should be Combine (“m” instead of “n”)
and so far.
In general, in my opinion the paper after indicated above corrections can be published.
Thank you.
Reviewer 2 Report
author has addressed all comments.
Author Response
Thank you for your guidance in the revision process of the paper.